# COVID-19 and the Change in Lifestyle: Bodyweight, Time Allocation, and Food Choices

**DOI:** 10.3390/ijerph181910552

**Published:** 2021-10-08

**Authors:** Xiaolei Li, Jian Li, Ping Qing, Wuyang Hu

**Affiliations:** 1College of Economics and Management, Huazhong Agricultural University, Wuhan 430070, China; lixiaolei120@webmail.hzau.edu.cn (X.L.); qingping@mail.hzau.edu.cn (P.Q.); 2Department of Agricultural, Environmental, and Development Economics, The Ohio State University, Columbus, OH 43210, USA; hu.1851@osu.edu

**Keywords:** COVID-19, lifestyle, bodyweight, time allocation, food choices

## Abstract

We analyze the dynamic changes in individuals’ lifestyle during the COVID-19 outbreak and recovery period through a survey of 1061 Chinese households. Specifically, we are interested in individuals’ bodyweight, time allocation and food choices. We find that COVID-19 is associated with weight gain, less time spent on exercise and more time on entertainment. The proportion of online food purchase and snack purchases also shows an upward trend. This study provides useful implications on the impact of COVID-19 and its associated lockdowns on individuals’ lifestyle and offers foresights for countries in different stages of the pandemic. It explains how encouraging exercise, managing new food purchase venues, and reducing the intake of unhealthy food such as snacks may also need to be considered in dealing with the aftermath of COVID-19.

## 1. Introduction and Literature Review

### 1.1. Introduction

According to the World Health Organization, COVID-19 has resulted in a total of approximately 179.241 million confirmed cases reported worldwide as of 23 June 2021, out of which approximately 3.890 million have died [1]. The outbreak of COVID-19 has had a significant impact on the world’s economic growth, trade, food supply, as well as individuals’ lifestyle [2,3,4,5,6]. In response to the global spread of COVID-19, the Chinese government has adopted various prevention and control measures, among which the lockdown policy effectively slowed down the spread of COVID-19. The high infectivity of the novel coronavirus is not only a potential health risk factor but can also have psychosocial impact and affect individuals’ lifestyle [7,8]. 

Three aspects of change in individuals’ lifestyle are analyzed in this paper: physical health, time allocation, and food choices. Firstly, previous studies have shown that human bodyweight is closely related to their physical health [9,10]. Therefore, we consider weight change as an important indicator to reflect the change of individuals’ physical health during different pandemic periods. Secondly, during the pandemic, individuals are restricted from leaving their homes by the lockdown policy, which may affect individuals’ time allocation [11,12]. Finally, the pandemic and its associated lockdowns may lead to changes in food purchasing habits and snacking frequency. Experts predict that online food purchase will soon challenge the status of the conventional food retail industry that has been dominating the retail market before the COVID-19 pandemic [13,14]. It is also evident that COVID-19 has led to a shift in the food supply and distribution model and eventually may also favor more online food retail [15]. The pandemic and its associated lockdowns have caused massive stress across the society. Ref. [16] shows that stress and anxiety lead to greater snack intake. 

We also note that the three aspects of lifestyle we study are related. The combined effect of fear of contracting the disease as well as the lockdown may reduce individuals’ time devoted to outdoor excise. At the same time, lack of exercise may lead to a vicious circle of weight gain [11]. Stress as well as more flexible time at home may all contribute to more snack intake and snacks may also lead to higher energy intake thus the risk of weight gain [17]. 

Many factors may contribute to the changes in individuals’ lifestyle during the pandemic. The outbreak and evolution of COVID-19 can change individuals’ risk perceptions. Even when the pandemic recedes somewhat, individuals may still worry about the possibility that the disease would break out again. Changing risk perception may further affect individuals’ lifestyle choices during the pandemic [18]. The role of social network cannot be ignored during the pandemic and the corresponding lockdowns. A social network is able to disseminate information about COVID-19 as well as allow individuals to perceive the world from others’ perspectives and obtain social and moral support from each other. 

This study makes three possible contributions: first, in previous research, there are few studies on individuals’ lifestyle choices that are directly related to COVID-19. Survey data and second-hand data are combined in this study to explore the impact of COVID-19 on individuals’ lifestyle. Second, since the pandemic has undergone rapid changes and China has gone through different stages related to the pandemic, this study compares two pandemic periods and analyzes whether they have different effects on individuals’ physical health, time allocation and food choices. We define two periods: COVID-19 outbreak period (January–March 2020) and COVID-19 recovery period (April–June 2020). Third, we explain the change of lifestyle by a series of subjective variables (risk aversion, fear of COVID-19 resurgence and size of social network) as well as objective variables (confirmed case of COVID-19, search frequency of COVID-19 related terms, as well as lockdown duration).

This study conducts an empirical analysis of the change of individuals’ lifestyle and associated factors during the COVID-19 pandemic. We refer to the first quarter of 2020 as the outbreak period in China and the second quarter as the recovery period. We provide evidence on three aspects of individuals’ lifestyle in the two periods: physical health, time allocation, and food choices.

### 1.2. Literature Review

Previous studies have analyzed the impact of COVID-19 and the subsequent lockdowns on the overall economy, food prices, supply chain, as well as food consumption [5,19,20,21]. Self-quarantine and social lockdowns can effectively limit the spread of the virus [22], but isolation may change individuals’ day to day lifestyle. Therefore, this study analyzed the impact of COVID-19 on individuals’ lifestyle from three aspects: physical health, time allocation and food choices. 

Changes in bodyweight tend to be correlated with individuals’ physical health [9,10]. Recent studies have explored the impact of COVID-19 lockdown policy on bodyweight, showing a general positive relationship [23,24,25,26]. The difference in our study is that we take into account the possible differential influence COVID-19 outbreak and recovery period may have on individuals’ weight change. Specifically, the COVID-19 outbreak period (January–March 2020) is associated with a high incidence of confirmed cases and most cities/towns in China experienced lockdowns with various lengths. The COVID-19 recovery period (April–June 2020) is the revival stage, during which confirmed cases decreased significantly. General Office of the State Council of China issued a notice in April 2020 largely allowing economic activities to return to normal (PRC, 2020) [27]. At the same time, daily data on confirmed cases of COVID-19 released by the same Office showed that the pandemic had subsided in April [28]. 

The change in individuals’ lifestyle may also be reflected in individuals’ time allocation. COVID-19 has affected individuals’ time spent at home in general, as well as time spent on exercise and sleep [11,12,21,29]. In addition, some recent studies analyze the change in individuals’ exercise time under the influence of COVID-19 from the perspective of behavioral theory [30]. In contrast, our study focused on individuals’ time allocation exemplified by both exercise and entertainment time during the pandemic. We investigated these in both the outbreak and recovery periods. 

The COVID-19 outbreak has led to significant changes in food consumption and production around the world [31,32]. The high infectivity of the virus raises consumer concerns about transmission of the virus through the food transportation channel [33]. The change in individuals’ lifestyle is also reflected in their food consumption habits [11,21,26,34,35]. Studies on online purchases of agricultural products find that the pandemic has redirected a large amount of consumer attention to online platforms [15]. Online food purchase reduces the exposure of consumers in physical retail stores, which may mitigate the risk of infection. On the other hand, COVID-19 may also affect food prices [20]. Disruption to the food supply chain as a result of the lockdowns is an important driver for food price increases [5,20]. As a result, this study took price into account when analyzing consumers’ online food purchase behavior during the pandemic. In addition, given the lockdowns and restrictions on social interaction during the pandemic, some studies find that stress or loneliness could lead to high intake of hedonic food and snacks [16,36,37]. However, few studies have analyzed whether COVID-19 has any effect on hedonic snack purchase. We filled this gap by investigating the dynamics of online food purchase and snack purchase during the COVID-19 outbreak and recovery period.

As such, the key variables this study is interested in are bodyweight, time allocation, online food purchase, as well as snack purchase. In order to examine the changes of these key variables we considered a series of covariates. In addition to impacts on human physical health, studies have shown that the pandemic may also trigger psychological or psychosocial reactions which may in turn affect their behavior [38]. We investigated the following aspects: *risk aversion, fear of resurgence* and *size of social network*.

From a public health standpoint, increased risk perception and risk aversion motivate individuals to adopt behaviors to reduce possible infection [18]. Ref. [39] designed a simple lottery experiment that could measure the degree of risk aversion over a wide range of payoffs. Ref. [40] extended the experiment in [39] to study famers’ risk perception and aversion towards new technologies. In our study, we adopted the method of [40] to measure *risk aversion*.

Perception on the risks associated with COVID-19 expresses an individual’s subjective assessment of the likelihood of COVID-19 infection. Ref. [41] analyzed individuals’ doubts, concerns and fears about COVID-19. Ref. [42] used a 10-country sample and showed that individuals have high level of concern about COVID-19 across geographic regions. Based on this, we asked respondents about their *fear of resurgence* of the pandemic in the future to reflect their risk perception and concern about COVID-19.

Social interactions, such as verbal communication and smartphone apps, could alleviate loneliness, improve mental health and affect individuals’ behavior [43,44,45]. However, few previous studies have taken social networks into research when studying behavior under the pandemic. During the COVID-19 outbreak period, individuals’ interaction with each other was limited. As a result, social media became the primary way for social connection. Therefore, we used the size of friends in social media to gauge the *size of social network* and include it as an independent variable. To sum up, we included subjective variables risk aversion, fear of resurgence and size of social network into the model. Although the size of social network is strictly an objective variable, its formation and scale are subject to each individual. As a result, we also grouped it as a subjective independent variable.

In terms of objective variables that are outside the control of individuals, the authors of [15] included COVID-19 confirmed cases per week and COVID-19 cumulative confirmed cases each week as an independent variable in their analysis on the impact of COVID-19 on online food purchases. We also considered a similar independent variable confirmed case to reflect the severity of the pandemic in each region of our data. Additionally, following [15], we took live internet search index to reflect individuals’ actual knowledge and concern on the pandemic in their local area. In China, Baidu is the most popular internet search engine [46]. Therefore, we used Baidu keyword search index to capture total search frequency of pandemic-related information search in each Chinese city during both periods, and we labelled this variable search frequency as an objective variable. In addition, we also included the length of lockdown in each region through variable lockdown duration.

To sum up, this paper studies the impact of COVID-19 on individuals’ lifestyle in terms of physical health, time allocation and food choices during the pandemic outbreak and recovery period in China. In this process, we considered both the subjective independent variables that may affect individuals’ psychological status and subsequently their behavior, as well as objective independent variables that may concern the surveyed individuals.

## 2. Methods and Material

### 2.1. Data Collection

We used an online survey to record participants’ lifestyle during the COVID-19 outbreak (January–March 2020) and recovery period (April–June 2020) through their recall. Samples were collected according to the population of each province in China’s Sixth Population Census. A total of 1106 observations were collected nationwide, of which 1061 were valid questionnaires, suggesting an effective rate of 95.93%. Meanwhile, our data were matched with Baidu search indexes measuring how “heated” the pandemic-related topics are among individuals in each Chinese city through their online search activities. In China, all land is administratively assigned to one of its cities. In this study, we define each city by its urban center as well as its associated regional districts, which usually include rural areas many times larger than its urban core. We provide a detailed explanation of this variable later in the variable measurement section. Furthermore, we obtained the number of confirmed COVID-19 cases in Chinese cities through GitHub (GitHub is the world’s largest open source database).

Our questionnaire design began in April 2020. We tested the survey questionnaire through five focus group discussions which were held every two weeks, with 3–5 members in each focus group. We corrected and modified the questions in the questionnaire according to comments from each focus interview. We completed the first draft of the questionnaire in mid-August 2020. Finally, we spent two weeks carrying out five rounds of online pre-survey tests to improve the questionnaire. After each round, minor formatting and wording changes were implemented in the questionnaire following the issues reflected in the pre-survey tests. We completed final data collection in early September 2020.

### 2.2. Variable Measurement

#### 2.2.1. Dependent Variable

We took weight change as an indicator for physical health, exercise time and entertainment time as indicators for time allocation, and the proportion of food online purchase and the proportion of snack purchases as indicators for food choices. 

We define variables Δ Weight t1 (January–March 2020) and Δ Weight t2 (April–June 2020) as the difference between the three-month mean bodyweight of the COVID-19 outbreak period and recovery period compared to the corresponding mean bodyweight of the same period of the previous year, respectively. It should be noted that in the questionnaire, we asked respondents to report their bodyweight as of 1 January 2020; 31 March 2020; 30 June 2020. Meanwhile, assuming that individuals’ bodyweight does not change too drastically given no major life-changing events affecting the entire population, we use data at a point in time to represent a period. We assume individuals’ bodyweight of 1 January 2020 can approximate that before the pandemic, that is, their weight of January–March 2019 and April–June 2019 is equal to the weight of 1 January 2020. Thus, the four periods of mean weight are January–March 2020; April–June 2020; January–March 2019; April–June 2019. As a result, Δ Weight t1 is the difference between mean bodyweight of March 2020 and mean bodyweight of January 2019 (represented by weight reported for January 2020), and Δ Weight t2 the difference between mean bodyweight of June 2020 and mean bodyweight of June 2019 (also represented by bodyweight reported for January 2020). 

Similarly, Δ Exercise time (t1 and t2) represents the difference between the three-month mean exercise time (per day) in January–March 2020 or April–June 2020 and those of the same period of previous year, respectively. Δ Entertainment time (t1 and t2) represents the difference between the three-month mean entertainment time (per day) in the two periods in 2020 and the corresponding two periods in 2019, respectively. Online food purchase is the proportion of food purchased online over all food purchased. We asked respondents to recall the three-month average amount of Online food purchase over each of the two periods in 2020 and the corresponding two periods in 2019. As a result, Δ Online food purchase (t1 and t2) represents the difference between the value of Online food purchase in 2020 and 2019 for the January–March and April–June periods, respectively. Snack purchase is defined as the proportion of snacks purchased as the percentage of total food purchased. Our questionnaire directly asks respondents the difference between their Snack purchase in 2020 and 2019 in the January–March and April–June period, respectively. This is different to how we measure the other lifestyle variables where we first acquire data of each of the four periods and then calculate the difference ex post. The reason is that focus groups respondents reported that directly considering the difference for Snack purchase is more helpful to improve the accuracy of their recalled answers. Δ Snack purchase (t1 and t2) represents the difference in Snack purchase between 2020 and 2019 for the January–March and April–June period, respectively. 

#### 2.2.2. Subjective Independent Variables

Risk aversion: our questionnaire followed the lottery experimental design of [38]. Relative risk aversion is obtained according to the results of a series of experiments. A higher value in measurement suggests more the risk averse. Fear of resurgence: in our questionnaire, we asked respondents “Do you think there will be a second outbreak? “The options are “yes”, “no” or “not sure”. In the process of data processing, we formed a dummy variable that is equal to one if the selected response is “yes” and combine the “no” and “not sure” responses into zero for the dummy variable. Size of Social Network: we asked respondents “How many friends do you believe you have on all social media (such as WeChat, Weibo, QQ)? (these are the three most popular social media apps in China by number of users) Please include only individual friends but exclude individuals in groups that you belong to, whom you have never spoken to in person?” Once again, since respondents were asked to add up the total number of friends they have, we allowed some level of subjectivity in this process.

#### 2.2.3. Objective Independent Variables

Confirmed case: GitHub reports city-level total number of cumulative confirmed cases of COVID-19 each day. We added the total number of confirmed cases for each of the outbreak and recovery period in each Chinese city. We then divided the number of confirmed cases in each city in our sample by their corresponding size of population. We conducted the same analysis for both the outbreak and recovery period. Search frequency: we obtained this variable that measures web search activities on Baidu.com from Baidu Trend Index [47]. This index is city-specific and is proportional to the number of raw searches individuals conduct through Baidu.com with a proprietary weighting scheme. We used this index to describe the frequency of web searches related to two keywords, “COVID-19” and “pandemic”, during the COVID-19 outbreak and recovery period, respectively. We then divided the total frequency of each city in each of the two periods by the total population of the corresponding city. Lockdown duration: during the pandemic, various cities and communities within cities have experienced lockdowns of different lengths. We asked respondents to report for how many weeks their residential community was under strict lockdown during the two periods, respectively.

#### 2.2.4. Other Independent Variables

Package delivery restriction: following a similar notion as the strict lockdown, some communities also enacted soft-enforced restrictions on their residents to send or retrieve delivered packages (individual-oriented package (express) delivery serve plays an integrated role in the daily lives of Chinese. Items delivered range from general consumer products to fresh grocery items, including pre-cooked meals and single-serving drinks. Based on data from the State Post Bureau of the People’s Republic of China (http://www.spb.gov/xw/dtxx_15079/, accessed on 6 May 2020), in 2020, Chinese received a total of 83.36 billion pieces of packages, or about 1.1 pieces per person per week) as a means to further limit social interaction. We asked respondents to report the duration of this restriction in both periods. Duration of COVID: respondents indicated for how many months they expected the pandemic were to last during the two periods. Experience starvation: this dummy variable measures whether respondents had ever suffered from involuntary starvation in their lifetime. Diagnosed is a dummy variable, indicating whether there were confirmed or highly suspected cases of infection for respondents themselves or among their families and social contacts. Stores nearby is a count variable showing the number of food markets that are within a 15 minutes walking or five minutes driving distance from the respondents’ home. Δ Price (t1 and t2): the difference between the three-month mean price of snacks purchased in January–March 2020 and the same period in 2019 (t1); as well as the difference between that in April–June 2020 and the same period in 2019 (t2), respectively.

### 2.3. Model

We used a province fixed effect model to analyze our lifestyle variables given the COVID-19 pandemic:(1)$$Y_{ijt}=\partial +\beta^{\prime }X_{ijt}+\gamma^{\prime }Z_{ijt}+\delta^{\prime }\varphi +C_{pro}+\varepsilon_{ijt}$$
where $Y_{ijt}$ is lifestyle variable *j* for individual  i during time t. In our context, these are measured by differences: Δ weight, Δ exercise time, Δ entertainment time, Δ online food purchase, Δ snack purchase. $\;X_{ijt}$ is a vector of independent variables that includes risk aversion, confirmed case, search frequency, lockdown duration, fear of resurgence and size of social network; $Z_{ijt}$ is the vector of other independent variables that includes package delivery restriction, experience starvation, duration of COVID, stores nearby and Δ Price; φ includes demographic variables such as women, age, and married; $\;C_{pro}$ is a province fixed effect; $\varepsilon_{ijt}$ is a random error.

We compared the models with and without province fixed effects and found the two sets of results qualitatively identical. We also considered the effect of interaction terms in our models. However, F tests suggest that these interaction terms are jointly insignificant in the model with or without province fixed effects. As such, we did not include these variables into the model. As shown in Appendix A, VIF values indicate that there are little concerns of collinearity between independent variables.

## 3. Results

### 3.1. Summary Statistics

Figure 1 reports the sample mean of individuals’ bodyweight, exercise time, entertainment time, the proportion of online food purchase, as well as the proportion of snack purchases during the two periods. In each panel of Figure 1, we report the sample average. In addition, we also show five provinces/cities with the most confirmed cases of COVID-19 including Beijing, Shanghai, Guangdong, Hubei and Chongqing (according to the statistics of COVID-19 confirmed cases in Chinese cities on GitHub, these five provinces or cities are with the highest number of COVID-19 confirmed cases during both periods. Among them, Beijing, Shanghai, and Chongqing are provincial-level cities meaning that their administrative privileges are equivalent to a province.). We refer to the columns under sample mean and each province/city in a figure as a cluster. The four bars in each cluster in Figure 1a–e represent the mean values of the variable in January–March 2020, April–June 2020, January–March 2019 and April–June 2019, respectively. In Figure 1e, since we directly obtained the difference in Snack purchase in the two 2020 periods compared to the same corresponding periods in 2019, only two bars are presented for under each cluster.

Figure 1 shows that the sample mean of bodyweight, entertainment time, the proportion of online food purchase, and the proportion of snack purchases during the outbreak period are all higher than their corresponding quantities during the recovery period (i.e., for each cluster, compare columns 1 and 3 in Figure 1a,c,d and compare columns 1 and 2 in each cluster of Figure 1e. On the other hand, exercise time (Figure 1b) has the opposite trend. From the perspective of different provinces and cities, only Chongqing showed a trend inconsistent with the sample mean in terms of bodyweight and the proportion of online food purchase. Comparing the COVID-19 outbreak period and recovery period with the corresponding period of the previous year (i.e., column 1 versus column 2, and column 3 versus column 4, respectively, in each cluster for Figure 1a–d and each column in each cluster of Figure 1e, the sample mean of bodyweight, entertainment, the proportion of online food purchase and the proportion of snack purchases are higher than those in the same period of previous year while exercise time is the opposite. This is true for both periods. From the perspective of different provinces and cities, the outcomes of exercise time in Guangdong province during the recovery period are inconsistent with the sample mean. In terms of snack purchases, Hubei and Chongqing show an inconsistent trend with the sample mean.

Table 1 reports sample descriptive statistics. The sample mean value of dependent variables also indicates that bodyweight, entertainment time, the proportion of online food purchase, and proportion of snack purchases during the two periods are all higher than the corresponding period of the previous year, while the exercise time is the opposite, which is consistent with the conclusion in Figure 1. Among the independent variables, the sample mean of risk aversion is 1.499. According to [39,40], the value indicates a trend towards risk neutrality. A total of 16% of respondents believed that the COVID-19 would break out again in 2020. The sample average of size of social network is 229 individuals. There were about 1.814 confirmed cases per 10,000 individuals during the outbreak period and it fell to 0.039 per 10,000 individuals during the recovery period, suggesting a significant decrease in confirmed cases in all regions of China. Variable search frequency dropped from 5.706 per 100 individuals during the outbreak period to 3.926 per 100 individuals during the recovery period. Lockdown duration was less during the recovery period than during the outbreak period. For other independent variables, we paid attention to prices. The respondents believed that the average food price went up by nearly 8% during the outbreak period compared to the same period in 2019. During the recovery period, prices went up by about 5%. For control variables, 48% of our respondents were female. Average respondent age was 33.89. We had disproportionally more respondents with college Education. The average annual household income was RMB 204,000. Both education and income were higher than the national average.

### 3.2. Empirical Results

Table 2 contains our province fixed effect estimates of bodyweight and time allocation equations and Table 3 contains results for food choices. Reflected by Table 2, fear of resurgence was positively correlated with bodyweight in both the outbreak and recovery period. The marginal effect of fear of resurgence increased from 0.801 kg in the outbreak period to 0.989 kg in the recovery period. At the same time, respondents with one unit (equals 100 individuals) more in the size of social network are associated with 0.205 kg more in bodyweight during the outbreak period but such association is insignificant in the recovery period. We found 0.066 kg increase in bodyweight in the outbreak period and 0.170 kg increase in the recovery period can be associated with lockdown duration increasing by one week. Variables risk aversion, confirmed case and search frequency were uncorrelated with bodyweight in both the outbreak and recovery period.

Risk aversion was only correlated with exercise time in the outbreak period. While the magnitude of the risk aversion scale may be less obvious to interpret, the direction is clear: exercise time increased corresponding to the decrease of risk aversion. One unit (equals 100 individuals) increase in the size of social network was related to 0.017 h per day reduction in exercise time during the outbreak period. When package delivery restriction increased by one week, exercise time was found to decrease by 0.013 h per day during the outbreak period. The number of food markets within 15 min of walking or five minutes of driving was related to resident exercise time, that is, with one more store nearby, exercise time was observed to increase by 0.015 h per day during the outbreak period. In addition, age, health status, family size, either child or elderly at home were related to exercise time in the two different periods. On the other hand, fear of resurgence, confirmed case, search frequency and lockdown duration were uncorrelated with exercise time in both periods.

Moving on to entertainment time, in the recovery period, individuals believing COVID-19 would outbreak again (variable fear of resurgence) were associated with 0.323 h increase in entertainment time per day. Increase in one unit (100 individuals) in size of social network was associated with 0.138 and 0.057 h increase in entertainment time per day in the two periods, respectively. With an increase of one more confirmed case per 10,000 individuals, entertainment time decreased 1.941 h per day in the recovery period. One more week of lockdown duration was related to 0.074 and 0.077 h of increase in entertainment time per day in the two periods, respectively. One more week of package delivery restriction was related to 0.107 h increase in entertainment time per day. Duration of COVID-19 increased by one month was related to entertainment time decreasing by 0.072 h per day in the outbreak period. Having experienced starvation (experience starvation) had negative correlation with entertainment time, reducing it by 0.452 and 0.239 h per day in the two periods, respectively. Moreover, household member a medical staff was negatively correlated with entertainment time only in the outbreak time. We found that risk aversion and search frequency were irrelevant to entertainment time in both periods.

Table 3 reports fixed effect estimates for food choices. We found that one unit (equals 100 individuals) increase in size of social network was associated with 0.526% more of online food purchase in the recovery period. One unit increase in search frequency was associated with 0.654% increase in online food purchase in the outbreak period. Moreover, if a household had a household member or individual in their friend circle diagnosed with COVID-19 (diagnosed), it was associated with 6.114% and 5.652% reduction in the proportion of online food purchase in the outbreak and recovery period, respectively. Older individuals were found to be more likely associated with purchasing food online during the outbreak period. Married individuals were negatively correlated with the proportion of online food purchase in the recovery period, and education was positively correlated with the proportion of online food purchase in both periods. The presence of either child or elderly at home was positively associated with the proportion of online food purchase in both periods. Variable risk aversion, fear of resurgence, confirmed case and lockdown duration were irrelevant to the proportion of online food purchase in both two periods.

In the outbreak period, those believing COVID-19 would outbreak again (fear of resurgence) were associated with 6.359% increase in the proportion of snack purchase over all food. Size of social network raised by one unit (100 individuals) was associated with the proportion of snack purchase increasing by 1.919% in the outbreak period. Package delivery restriction increased by one week was related to 0.729% reduction in the proportion of snack purchase in the outbreak period. In the recovery period, if duration of COVID-19 increased by one month, the proportion of snack purchase was observed to reduce by 0.621%. Δ price increasing by 1% was associated with the increase in proportion of snack purchase by 0.312%. In addition, age was negatively correlated with the proportion of snack purchase in both periods. Married was negatively correlated with the proportion of snack purchase in the outbreak period. Finally, we found risk aversion, confirmed case, search frequency and lockdown duration were uncorrelated with the proportion of snack purchase in both periods.

## 4. Discussion

This study aimed to analyze how individuals’ lifestyle may change during the COVID-19 pandemic, relying on a survey in China. We find that COVID-19 has likely changed individuals’ lifestyle at least in terms of their physical health, time allocation and food choices. As the pandemic wanes, the impact of COVID-19 on individuals’ lifestyle might have diminished but has not completely disappeared compared to the same period of the previous year. In the premise of overweight being a common problem globally, individuals’ weight gain is another manifestation of the negative impact of COVID-19 on the society. Quaresma et al. (2021) and Wang et al. (2021) also show that COVID-19 causes weight gain in residents [21,26]. The pandemic has affected residents’ time allocation at home [11,12,21,29]. Reduced time in exercise and increased engagement in entertainment and snack purchase related to the pandemic may also exacerbate the negative impact. Quaresma et al. (2021) also state that the negative emotions caused by COVID-19 will increase the consumption of snacks [21]. COVID-19 has also changed residents’ shopping patterns, with an increasing proportion of online purchase [15]. In addition, psychological emotions, social relationships and lockdown policies are also likely factors related to lifestyle changes. Finally, although the instant online search index we considered did not seem to matter for bodyweight and time allocation, we do have moderate evidence that it can be related to food choices.

Cross-sectional data were used in this analysis to compare samples in the COVID-19 outbreak period and recovery period. Although our study could reflect the differences in associated factors in the two periods, we could not formally infer a causal relationship between variables. This remains to be a useful topic for future research.

## 5. Conclusions

Our research indicates that COVID-19 as well as lockdowns and self-quarantine triggered by it may lead to weight gain, less exercise time, more entertainment time, and greater online food purchase and snack purchase. Therefore, from a public health standpoint, governments could consider to raise individuals’ awareness of the pandemic’s potential detriment to health in addition to the danger of the virus itself. Encouraging the population to proactively allocate their personal time between exercise and entertainment might be useful. Individuals can be reminded to become more intentional on their meal planning and adjust their food choices to reduce the intake of unhealthy food such as snacks. In addition, this study finds that COVID-19 might have enhanced online food purchase, which may further contribute to the structural adjustment of the food retail industry [15]. This could not be ignored in future studies of the long-term impact of the pandemic and may have implications for other retail industries beyond food.

China is a developing country with a large population as well as complex cultural and social construct. As China is one of the few countries in the world that has largely gone through both the outbreak period and the recovery period, this research based on China may also provide insights and policy suggestions for other countries with similar conditions.

## Figures and Tables

**Figure 1 ijerph-18-10552-f001:**
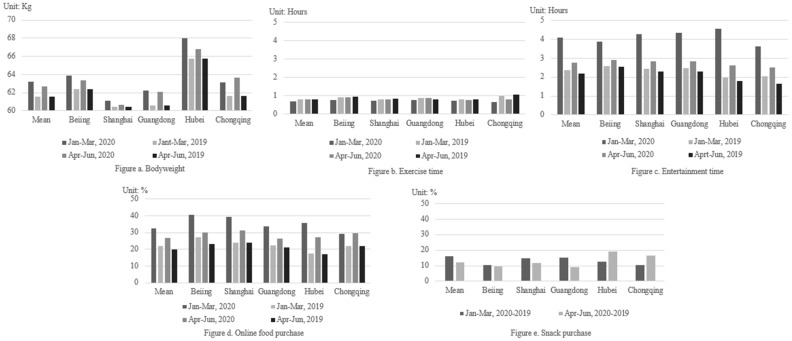
Trend in bodyweight (**a**), exercise time (**b**), entertainment time (**c**), online food purchase (**d**) and snack purchase (**e**) during COVID-19.

**Table 1 ijerph-18-10552-t001:** Descriptive statistics.

Variable	Mean	Std. Dev
**Dependent variable**		
Δ Weight t1 (kg) ^#^	1.759	3.168
Δ Weight t2 (kg) ^#^	1.246	3.130
Δ Exercise time t1 (h)	−0.121	0.547
Δ Exercise time t2 (h)	−0.017	0.430
Δ Entertainment time t1 (h)	1.735	2.030
Δ Entertainment time t2 (h)	0.585	1.194
Δ Online food purchase t1 (%)	10.630	22.120
Δ Online food purchase t2 (%)	6.966	15.970
Δ Snack purchase t1 (%)	16.140	35.050
Δ Snack purchase t2 (%)	12.300	28.140
**Main independent variable**		
Risk aversion (index)	1.499	1.273
Fear of resurgence (dummy)	0.160	0.367
Size of social network (per 100)	2.291	1.907
Confirmed case t1 (per 10,000)	1.814	9.346
Confirmed case t2 (per 10,000)	0.039	0.083
Search frequency t1 (times per 100 individuals)	5.706	2.724
Search frequency t2 (times per 100 individuals)	3.926	1.912
Lockdown duration t1 (# of weeks)	2.902	3.381
Lockdown duration t2 (# of weeks)	2.416	3.450
**Other independent variable**		
Package delivery restriction t1 (# of weeks)	0.567	1.482
Package delivery restriction t2 (# of weeks)	0.369	1.274
Duration of COVID-19 t1 (# of month)	3.760	2.413
Duration of COVID-19 t2 (# of month)	3.757	2.749
Experienced starvation (dummy)	0.152	0.359
Diagnosed (dummy)	0.078	0.269
Stores nearby (quantity)	3.718	2.751
Δ Price t1 (%)	7.994	9.629
Δ Price t2 (%)	5.083	8.634
**Control variable**		
Women (dummy)	0.480	0.500
Age (years)	33.890	7.409
Married (dummy)	0.221	0.415
Education ^$^ (categorical)	5.832	0.634
Health status ^	1.945	0.698
Income (annual pre−tax; per 10,000)	20.400	13.340
Family size (# of individuals)	3.299	1.034
Either child or elderly at home (dummy)	0.769	0.422
Household member a medical staff (dummy)	0.084	0.277

Note: ^#^ Indicators t1 and t2, respectively, correspond to the COVID-19 outbreak period, represented by January to March and the COVID-19 recovery period, represented by April to June. Following this definition, variable “Δ Weight t1” is defined as (bodyweight in t1 2020—bodyweight in t1 2019), whereas “Δ Weight t2” is defined as (bodyweight in t2 2020—bodyweight in t2 2019). Variables Δ Exercise time (t1 and t2) and Δ Entertainment time (t1 and t2) are similarly defined. For Δ Online food purchase, for each t1 and t2, it is defined as the percentage of total food purchased online and the difference is obtained similarly to the first three variables. Variable Δ Snack purchase (t1 and t2) is measured directly through the survey on the amount of snack purchased as the percentage of total amount of food purchased. ^$^ Education is measured with seven levels: 1 for uncompleted primary school, 2 for primary school graduate, 3 for middle high school graduate, 4 for high school (technical school/vocational high school) graduate, 5 for college graduate, 6 for university graduate, and 7 for Master or above graduate. ^ Health status is self-reported by respondents using one of the five levels: 1 for very good, 2 for good, 3 for fair, 4 for poor, 5 for very poor.

**Table 2 ijerph-18-10552-t002:** Empirical results for bodyweights and time allocation.

Variable	Δ Weight t1 ^#^	Δ Weight t2 ^#^	Δ Exercise Time t1	Δ Exercise Time t2	Δ Entertainment Time t1	Δ Entertainment Time t2
Risk aversion	−0.001	0.030	−0.034 **	−0.008	−0.023	−0.029
	(0.074)	(0.078)	(0.014)	(0.011)	(0.049)	(0.027)
Fear of resurgence	0.801 ***	0.989 ***	−0.025	−0.020	−0.098	0.323 ***
	(0.297)	(0.304)	(0.051)	(0.041)	(0.172)	(0.114)
Size of social network	0.205 ***	0.089	−0.017 **	0.003	0.138 ***	0.057 ***
	(0.063)	(0.066)	(0.009)	(0.007)	(0.037)	(0.020)
Confirmed case	−0.008	0.798	0.002	−0.533	0.004	−1.941 **
	(0.022)	(2.314)	(0.004)	(0.481)	(0.011)	(0.898)
Search frequency	0.016	0.041	−0.002	0.004	−0.014	−0.007
	(0.056)	(0.072)	(0.010)	(0.013)	(0.034)	(0.031)
Lockdown duration	0.066 **	0.170 *	−0.001	−0.008	0.074 ***	0.077 **
	(0.032)	(0.094)	(0.006)	(0.012)	(0.022)	(0.035)
Package delivery restriction			−0.013 **	−0.001	0.107 ***	0.036
			(0.006)	(0.012)	(0.023)	(0.040)
Duration of COVID-19					−0.072 ***	0.005
					(0.026)	(0.014)
Experience starvation					−0.452 ***	−0.239 **
					(0.169)	(0.113)
Stores nearby			0.015 **	0.005	0.038	−0.011
			(0.007)	(0.004)	(0.025)	(0.014)
Women	0.297	0.268	−0.012	−0.007	−0.033	0.013
	(0.199)	(0.202)	(0.034)	(0.027)	(0.125)	(0.075)
Age	0.003	−0.014	−0.005 **	−0.003	−0.012	−0.006
	(0.017)	(0.017)	(0.003)	(0.002)	(0.009)	(0.005)
Married	−0.304	−0.186	−0.033	−0.032	−0.162	0.017
	(0.261)	(0.300)	(0.049)	(0.037)	(0.182)	(0.105)
Education	0.029	0.209	−0.019	−0.005	−0.149	−0.086
	(0.161)	(0.172)	(0.032)	(0.024)	(0.113)	(0.075)
Health status	0.053	0.097	0.055 **	−0.004	0.113	0.007
	(0.154)	(0.157)	(0.026)	(0.020)	(0.096)	(0.052)
Income	−0.007	−0.018 *	0.002 *	−0.000	0.005	0.004
	(0.009)	(0.010)	(0.001)	(0.001)	(0.005)	(0.004)
Family size	0.050	0.090	−0.034	−0.036 **	0.045	0.085 *
	(0.115)	(0.117)	(0.021)	(0.017)	(0.064)	(0.044)
Either child or elderly at home	0.290	0.176	0.058	0.089 **	−0.119	0.057
	(0.260)	(0.284)	(0.051)	(0.040)	(0.186)	(0.106)
Household member a medical staff	0.281	0.230	0.015	0.082	−0.459 **	−0.133
	(0.415)	(0.390)	(0.069)	(0.063)	(0.212)	(0.146)
Constant	−0.181	−1.025	0.160	0.259	1.961 *	1.144 *
	(1.549)	(1.719)	(0.270)	(0.245)	(1.031)	(0.659)
Province FE	YES	YES	YES	YES	YES	YES
Observations	1061	1061	1061	1061	1061	1061
R-squared	0.064	0.055	0.057	0.043	0.139	0.085

Note: Robust standard errors in parentheses and models include province fixed effects. ***, **, * indicate significance at the 1%, 5% and 10% levels, respectively. ^#^ t1 is the COVID-19 outbreak period, represented by January–March 2020; t2 is the COVID-19 recovery period, represented by April–June 2020.

**Table 3 ijerph-18-10552-t003:** Empirical results for food choices.

Variable	Δ Online Food Purchase t1 ^#^	Δ Online Food Purchase t2 ^#^	Δ Snack Purchase t1	Δ Snack Purchase t2
Risk aversion	−0.107	0.185	0.773	0.878
	(0.499)	(0.362)	(0.856)	(0.737)
Fear of resurgence	0.102	−0.069	6.359 **	0.422
	(1.786)	(1.311)	(3.116)	(2.243)
Size of social network	0.166	0.526 *	1.919 ***	0.763
	(0.396)	(0.276)	(0.664)	(0.492)
Confirmed case	−0.140	−10.088	−0.169	−14.253
	(0.250)	(17.724)	(0.311)	(35.385)
Search frequency	0.654 *	0.098	0.591	0.555
	(0.392)	(0.430)	(0.584)	(0.804)
Lockdown duration	0.189	0.099	−0.063	−0.360
	(0.241)	(0.492)	(0.361)	(0.701)
Package delivery restriction	−0.249	0.023	−0.729 **	−0.793
	(0.259)	(0.565)	(0.351)	(0.937)
Duration of COVID-19			0.075	−0.621 **
			(0.460)	(0.308)
Diagnosed	−6.114 **	−5.652 **		
	(2.547)	(2.269)		
Δ Price			0.198	0.312 **
			(0.134)	(0.136)
Women	2.187	0.890	−1.611	0.618
	(1.379)	(0.978)	(2.116)	(1.779)
Age	0.174 *	0.024	−0.262 *	−0.263 **
	(0.104)	(0.074)	(0.143)	(0.120)
Married	−2.164	−2.658 **	−7.025 **	−3.335
	(1.955)	(1.306)	(3.043)	(2.396)
Education	4.379 ***	1.685 *	−1.033	−1.748
	(1.229)	(0.947)	(1.939)	(1.838)
Health status	−0.194	0.053	−0.019	0.266
	(1.024)	(0.753)	(1.569)	(1.310)
Income	0.059	0.019	0.055	−0.008
	(0.052)	(0.040)	(0.099)	(0.065)
Family size	−1.222	0.181	−0.856	0.234
	(0.838)	(0.544)	(1.189)	(1.013)
Either child or elderly at home	5.746 ***	3.624 ***	4.727	3.737
	(1.871)	(1.295)	(3.186)	(2.505)
Household member a medical staff	−1.606	−0.708	6.763	4.757
	(2.577)	(1.805)	(4.394)	(3.882)
Constant	−27.464 **	−7.370	11.423	20.358
	(11.311)	(9.321)	(17.148)	(17.811)
Province FE	YES	YES	YES	YES
Observations	1061	1061	1061	1061
R-squared	0.100	0.066	0.079	0.060

Note: Robust standard errors in parentheses and models include province fixed effects. ***, **, * indicate significance at the 1%, 5% and 10% levels, respectively. ^#^ t1 is the COVID-19 outbreak period, represented by January–March 2020; t2 is the COVID-19 recovery period, represented by April–June 2020.

## Data Availability

The datasets used and analyzed during the current study are available from the corresponding author on reasonable request.

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
