# Peer review of "COVID-19 and the Change in Lifestyle: Bodyweight, Time Allocation, and Food Choices"

_ijerph, 2021, doi:10.3390/ijerph181910552_

Round 1

Reviewer 1 Report

  1. I suggest to add the definitions(ex. incidence, the number of confirmed cases/a day and so on) the reason why was named April-June, 2020 to  COVID-19 recovery period and January-March, 2020 to COVID-19 outbreak period.
  2.  In '2. Literature review(p.2)', if the literature review was not conducted systematically like the PRISMA(Preferred Reporting Items for Systematic reviews and Meta-Analysis) checklist, '2. Literature review' content would be better to supplement the introduction or the discussion. If the literature review was conducted systematically, please state the process of literature review in detail. 
  3. Please present the focus group discussion process in detail, such as the number of people, period, and frequency. And it needs to describe how 'focus group discussion' was reflected in the design of the questionnaire.
  4. I suggest to present statistic data of correlation or multicollinearity between independent variables.
  5. Please describe the limitations of the study, such as not proving causal relationship because of cross-sectional study.

Author Response

Reply to comments made by Reviewer 1 on manuscript IJERPH -1378882 titled “COVID-19 and the change in lifestyle: Bodyweight, time allocation, and food choices.”

Thank you for providing useful comments on the earlier draft of our paper. We have revised the paper in response to your suggestions. We hope that the paper will now be seen as making an important contribution to the literature.

Comment 1: I suggest to add the definitions(ex. incidence, the number of confirmed cases/a day and so on) the reason why was named April-June, 2020 to COVID-19 recovery period and January-March, 2020 to COVID-19 outbreak period.

Response 1: We agree with the reviewer that we need to add the definitions the reason why we named April-June, 2020 as COVID-19 recovery period and January-March, 2020 as COVID-19 outbreak period.

Following your suggestion, in the updated second paragraph of Section 1.2 “Literature review” (on p. 2) we now write: “Specifically, the COVID-19 outbreak period (January-March, 2020) is associated with a high incidence of confirmed cases and most cities/towns in China experienced lockdowns with various length. The COVID-19 recovery period (April-June, 2020) is the revival stage, during which confirmed cases decreased significantly. General Office of the State Council of China issued a notice in April 2020 largely allowing economic activities to return to normal (PRC, 2020) [27]. At the same time, daily data on confirmed cases of COVID-19 released by the same Office showed that the pandemic had subsided in April [28]. In addition, according to GitHub data analysis, there were 19,155 confirmed cases per day during the COVID-19 outbreak period and only 526 confirmed cases per day during the COVID-19 recovery period.”

Comment 2: In '2. Literature review(p.2)', if the literature review was not conducted systematically like the PRISMA(Preferred Reporting Items for Systematic reviews and Meta-Analysis) checklist, '2. Literature review' content would be better to supplement the introduction or the discussion. If the literature review was conducted systematically, please state the process of literature review in detail. 

Response 2: Thank you for your suggestion. First, we would like to explain a bit about the study approach in our “Literature review” Section. As pointed out by the reviewer, our literature review did not adopt the approach of a Meta-analysis, but we did use a systematic approach. We took multiple approaches to compile this section, including a careful literature search using major economic literature databases and from the cited literature in the studies we have examined from these databases.

Second, according to your suggestion, we have modified the structure of the manuscript by combining “Literature review” and “Introduction”. Some content in previous “Literature review” has been moved to supplement the “Discussion”. We hope, through our modification, the manuscript could better fit the journal’s theme and structure requirements.

Comment 3: Please present the focus group discussion process in detail, such as the number of people, period, and frequency. And it needs to describe how 'focus group discussion' was reflected in the design of the questionnaire.

Response 3: Thank you for your useful comment. Following your suggestion, we have expanded the focus group discussion in detail.

In the updated second paragraph of Section 2.1 “Data collection”, we have revised the writing as follows (on p. 4): “Our questionnaire design began in April 2020. We tested the survey questionnaire through five focus group discussions, which were held every two weeks, with 3-5 members in each focus group. We corrected and modified the questions in the questionnaire according to comments from each focus interview. We completed the first draft of the questionnaire in mid-August 2020. Finally, we spent two weeks carrying out five rounds of online pre-survey tests to improve the questionnaire. After each round, minor formatting and wording changes were implemented in the questionnaire following the issues reflected in the pre-survey tests. We completed final data collection in early September 2020.”

Comment 4: I suggest to present statistic data of correlation or multicollinearity between independent variables.

Response 4: We agree with the reviewer that we need to add statistic data of correlation or multicollinearity between independent variables.

Following your suggestion, we have added a description of collinearity in the second paragraph of section 2.3 “Model” (on p. 6): “As shown in table A1 and A2, VIF values indicate that there is little concern of collinearity between independent variables.”

We have included table A1 and A2 in the appendix of the manuscript (the last page of the manuscript).

Table A1. VIF value for Bodyweights and Time Allocation

Variable

Δ Weight t1

Δ Weight t2

Δ Exercise time t1

Δ Exercise time t2

Δ Entertainment time t1

Δ Entertainment time t2

Risk aversion

1.05

1.05

1.06

1.05

1.06

1.06

Fear of resurgence

1.06

1.06

1.07

1.07

1.08

1.08

Size of social network

1.23

1.21

1.24

1.22

1.24

1.22

Confirmed case

4.85

5.47

4.86

5.48

4.88

5.48

Search frequency

2.45

2.53

2.46

2.53

2.46

2.54

Lockdown duration

1.11

1.10

1.45

1.46

1.45

1.47

Package delivery restriction

1.41

1.41

1.41

1.41

Duration of COVID

1.06

1.07

Experience starvation

1.11

1.12

Stores nearby

1.08

1.08

1.08

1.08

Women

1.09

1.09

1.09

1.09

1.10

1.09

Age

1.42

1.41

1.43

1.41

1.44

1.42

Married

1.55

1.55

1.56

1.55

1.56

1.55

Education

1.27

1.27

1.28

1.28

1.28

1.28

Health status

1.14

1.14

1.14

1.14

1.14

1.14

Income

1.26

1.26

1.26

1.26

1.28

1.28

Family size

1.39

1.40

1.40

1.40

1.40

1.40

Either child or elderly at home

1.61

1.61

1.62

1.62

1.62

1.62

Household member a medical staff

1.09

1.09

1.09

1.09

1.13

1.13

Table A2. VIF value for Food Choices

Variable

Δ Online food

Purchase t1

Δ Online food

Purchase t2

Δ Snack

 Purchase t1

Δ Snack

Purchase t2

Risk aversion

1.05

1.05

1.05

1.05

Fear of resurgence

1.07

1.07

1.08

1.08

Size of social network

1.24

1.22

1.23

1.22

Confirmed case

4.89

5.50

4.87

5.47

Search frequency

2.46

2.53

2.46

2.53

Lockdown duration

1.45

1.45

1.46

1.47

Package delivery restriction

1.41

1.41

1.41

1.41

Duration of COVID

1.06

1.07

Diagnosed

1.15

1.16

Δ Price

1.25

1.12

Women

1.09

1.09

1.09

1.09

Age

1.43

1.42

1.44

1.42

Married

1.55

1.55

1.56

1.55

Education

1.28

1.28

1.28

1.28

Health status

1.14

1.14

1.14

1.14

Income

1.26

1.26

1.26

1.26

Family size

1.40

1.40

1.41

1.40

Either child or elderly at home

1.62

1.62

1.62

1.62

Household member a medical staff

1.17

1.17

1.12

1.12

Comment 5: Please describe the limitations of the study, such as not proving causal relationship because of cross-sectional study.

Response 5: Thank you for this comment. We agree with the reviewer that we need to add discussions on the limitations of the paper.

In the updated last paragraph of Section 4 “Discussion”, we wrote (on p. 13): “Cross-sectional data were used in this analysis to compare samples in the COVID-19 outbreak period and recovery period. Although our study could reflect the differences in the correlation of variables in the two periods, we could not formally infer causal relationship between variables. This remains to be a useful topic for future research.”

We thank you again for your valuable comments!

Reviewer 2 Report

Dear Authors

The study is interesting, especially because of the mathematical formulas used. The results may be of use to other researchers. However, I have technical and substantive reservations about the manuscript.

Technical Notes

  The structure of the manusccript should be in line with the template, i.e. there should be chapters: Introduction, Methods and Material (Subjects) (here you should include all the data on on-line research, time, people, mathematical formulas, etc., and statistical analyzes), Results (from short description of tables), Discussion, Conclusions.

Please read the reference requirements carefully, there should be an abbreviation of the journal name, the year should be bolded and the volume in italics.

It is not advisable to explain the cited literature sources in the footer of the manuscript, it should be included in the References.

Substantive comments

In my opinion, the content of the Introduction and Literature Review should constitute one chapter entitled Admission. It should justify the research and briefly present the state of knowledge on the topic under study. Fragments regarding the purpose of the research, e.g. lines 29-30, 37-38, 45-46 and similar, should be collected at the end of the Introduction. It would be good to end with a research hypothesis.

Many excerpts from the Literature Review chapter can be used in formulating the strengths of the research and in the Discussion of the results. In general, various parts of the Literature Review should be transferred to other chapters.

Chapter 3 should be Methods and Material, divided into appropriate subsections.

Subsection 3.2. The statistic summary should be called Results.

The description of the results is correct, but you should keep such a rule that the description of a given table should always precede it.

Chapter 4. Regression analysis should be subsection Results, section 4.1. The model should be included in the method and material.

After the Results there should be a Discussion, and only at the end Conclusions.

In the discussion, the obtained observations should be discussed and explained using the literature.

In the section Conclusions, the results are re-described, that is, incorrectly. As conclusions can be used subsection 5.2, and at its end give implications.

In conclusion, the content has a scientific value, but I have reservations about the way it was prepared.

In my opinion, the manuscript needs to be completely rewritten.

Please check out the other articles in IJERPH.

Author Response

Reply to comments made by Reviewer 2 on manuscript IJERPH -1378882 titled “COVID-19 and the change in lifestyle: Bodyweight, time allocation, and food choices.”

Thank you for providing useful comments on the earlier draft of our paper. We have revised the paper in response to your suggestions. Specifically, we modified the overall structure of the manuscript. We hope that the paper will now be seen as making an important contribution to the literature.

Comment 1: The structure of the manuscript should be in line with the template, i.e. there should be chapters: Introduction, Methods and Material (Subjects) (here you should include all the data on on-line research, time, people, mathematical formulas, etc., and statistical analyzes), Results (from short description of tables), Discussion, Conclusions.

Response 1: Thank you so much for providing valuable suggestions on our manuscript, we have significantly revised and adjusted the structure of the manuscript according to the IJERPH’s template and your suggestions. We have made our efforts to improve the structure and logic of the manuscript. We hope these improvements will make our paper suitable for publication in International Journal of Environmental Research and Public Health.

Comment 2: Please read the reference requirements carefully, there should be an abbreviation of the journal name, the year should be bolded and the volume in italics.

Response 2: Thank you for your comment, we have carefully modified the format of the reference according to the requirements of the IJERPH.

Comment 3: It is not advisable to explain the cited literature sources in the footer of the manuscript, it should be included in the References.

Response 3: Thank you for your comment. We have moved citations from the footer to the reference list.

Comment 4: In my opinion, the content of the Introduction and Literature Review should constitute one chapter entitled Admission. It should justify the research and briefly present the state of knowledge on the topic under study. Fragments regarding the purpose of the research, e.g. lines 29-30, 37-38, 45-46 and similar, should be collected at the end of the Introduction. It would be good to end with a research hypothesis.

Response 4: Thank you for the good suggestion. First, we have revised the manuscript strictly according to template, and combined the “Introduction” and “Literature review” section. In the combined section (“Introduction and literature review” section), we have discussed the state of knowledge on the topic under study. Second, following your comment, we have modified the research purposes of the manuscript. We have deleted lines 31-33,40-41,48-50,65-67, which can be seen in the revised version. In addition, we have put the research purpose at the end of the “Introduction” section. The details are as follows(on p. 2):

“This study conducts an empirical analysis of the change of individuals’ lifestyle and associated factors during the COVID-19 pandemic. We refer to the first quarter of 2020 as the outbreak period in China and the second quarter as the recovery period. We provide evidence on three aspects of individuals’ lifestyle in the two periods: physical health, time allocation, and food choices.”

Comment 5: Many excerpts from the Literature Review chapter can be used in formulating the strengths of the research and in the Discussion of the results. In general, various parts of the Literature Review should be transferred to other chapters.

Response 5: Thank you for the suggestion. According to your suggestion, we have modified the structure of the manuscript by combining “Literature review” and “Introduction”. Some content in previous “Literature review” has been moved to supplement the discussion. We hope,  through our modification, the manuscript is now a better fit for the IJERPH’s theme and structure requirements.

Comment 6: (1)Chapter 3 should be Methods and Material, divided into appropriate subsections. (2)Subsection 3.2. The statistic summary should be called Results. (3) The description of the results is correct, but you should keep such a rule that the description of a given table should always precede it. (4) Chapter 4. Regression analysis should be subsection Results, section 4.1. The model should be included in the method and material . (5)After the Results there should be a Discussion, and only at the end Conclusions.

Response 6: Thank you for the useful comment. We agree with the reviewer and have revised the overall framework of the manuscript. After modification, the logic flow across sections of the manuscript is more natural.

Comment 7: In the discussion, the obtained observations should be discussed and explained using the literature.

Response 7: Thank you for the valuable comment. We agree with the reviewer that the obtained observations could be discussed and explained by comparing with existing literature. Therefore, in the “Discussion” Section, we have added more citations to better explain our viewpoints. At the same time, we have added the limitation to the manuscript based on the review comment. The revised content of “Discussion” Section is as follows (on p. 12):

“This study aims to analyze how individuals’ lifestyle may change during the COVID-19 pandemic, relying on a survey in China. We find that COVID-19 has likely changed individuals’ lifestyle at least in terms of their physical health, time allocation and food choices. As the pandemic wanes, the impact of COVID-19 on individuals’ lifestyle might have diminished, but have not completely disappeared compared to the same period of the previous year. In the premise of overweight being a common problem globally, individuals’ weight gain is another manifestation of the negative impact of COVID-19 on the society. Quaresma et al(2021) and Wang et al(2021) also show that COVID-19 causes weight gain in residents[21,26]. The pandemic has affected residents’ time allocation at home [11,12,21,29]. Reduced time in exercise and increased engagement in entertainment and snack purchase related to the pandemic may also exacerbate the negative impact. Quaresma et al(2021) also states that the negative emotions caused by COVID-19 will increase the consumption of snacks[21]. COVID-19 has also changed residents’ shopping patterns, with an increasing proportion of online purchase [15]. In addition, psychological emotions, social relationships and lockdown policies are also likely factors related to lifestyle changes. Finally, although the instant online search index we considered did not seem to matter for bodyweight and time allocation, we do have moderate evidence that it can be related to food choices.

Cross-sectional data were used in this analysis to compare samples in the COVID-19 outbreak period and recovery period. Although our study could reflect the differences in associated factors in the two periods, we could not formally infer causal relationship between variables. This remains to be a useful topic for future research.”

Comment 8: In conclusion, the content has a scientific value, but I have reservations about the way it was prepared.

Response 8: We have taken your comments very seriously, and have systematically and carefully revised the manuscript. We hope that, following your guidance, our manuscript’s structure and logic are clearer.

Comment 9: In my opinion, the manuscript needs to be completely rewritten.

Response 9: Thank you for your valuable suggestions. According to your comments, our manuscript has been completely rewritten. We hope the modifications will make our paper suitable for publication in IJERPH.

We thank you again for your valuable comments!

Round 2

Reviewer 2 Report

Dear Authors
it's nice that you took my comments into account, but one more technical thing remains. In the previous review, I gave a note that was not used - the rule should be that the description of a given figure or table should always precede it. Thus the description of Figure 1 should start at 3.1 Summary Statistics and be before Figure.
Likewise, the description of Table 1 should be before the table, etc.
Please edit it.

Author Response

Reply to comments made by Reviewer 2 on manuscript IJERPH -1378882 titled “COVID-19 and the change in lifestyle: Bodyweight, time allocation, and food choices.”

Thank you for providing useful comments on the earlier draft of our paper. We have revised the paper in response to your suggestions. We hope that the paper will now be seen as making an important contribution to the literature.

Comment 1: it's nice that you took my comments into account, but one more technical thing remains. In the previous review, I gave a note that was not used - the rule should be that the description of a given figure or table should always precede it. Thus the description of Figure 1 should start at 3.1 Summary Statistics and be before Figure. Likewise, the description of Table 1 should be before the table, etc. Please edit it.

Response 1: Thank you for providing this useful comment on our manuscript. We are sorry that we misunderstood your comment in the previous round of revision. According to your comment, we have adjusted the position of the figure and tables in the manuscript to ensure that their description goes before the actual figure and table. We hope these improvements will make our paper suitable for publication in International Journal of Environmental Research and Public Health.

We thank you again for your valuable comment!
